# Important Role and Properties of Granular Nanocellulose Particles in an In Vitro Simulated Gastrointestinal System and in Lipid Digestibility and Permeability

**DOI:** 10.3390/biom13101479

**Published:** 2023-10-03

**Authors:** Warathorn Chumchoochart, Nopakarn Chandet, Chalermpong Saenjum, Jidapha Tinoi

**Affiliations:** 1Interdisciplinary Program in Biotechnology, Graduate School, Chiang Mai University, Chiang Mai 50200, Thailand; warathorn_c@cmu.ac.th; 2Department of Chemistry, Faculty of Science, Chiang Mai University, Chiang Mai 50200, Thailand; nopakarn_c@cmu.ac.th; 3Department of Pharmaceutical Sciences, Faculty of Pharmacy, Chiang Mai University, Chiang Mai 50200, Thailand; chalermpong_s@cmu.ac.th; 4Center of Excellence in Materials Science and Technology, Chiang Mai University, Chiang Mai 50200, Thailand

**Keywords:** emulsion, free fatty acid, gastrointestinal tract, granular nanocellulose, in vitro digestion, lipid digestibility

## Abstract

This research evaluated the role and feasibility of the granular nanocellulose particles (GNC) from sugarcane bagasse obtained from enzymatic hydrolysis in reducing lipid digestibility and permeability in an in vitro simulated gastrointestinal (GI) system. GNC concentration (0.02%, *w*/*v*) had significantly affected the released free fatty acids (FFA), with a reduction of approximately 20%. Pickering emulsion of a GNC and olive oil simulation mixture revealed higher oil droplet size distribution and stability in the initial stage than the vortexed mixture formation. The difference in particle size distribution and zeta potential of the ingested GNC suspension and GNC–olive oil emulsion were displayed during the in vitro gastrointestinal simulation. GNC particles interacted and distributed surrounding the oil droplet, leading to interfacial emulsion. The GNC concentration (0.01–0.10%, *w*/*v*) showed low toxicity on HIEC-6 cells, ranging from 80.0 to 99% of cell viability. The release of FFA containing the ingested GNC suspension and GNC–olive oil emulsion had about a 30% reduction compared to that without the GNC digestion solution. The FFA and triglyceride permeability through the HIEC-6 intestinal epithelium monolayer were deceased in the digesta containing the ingested GNC and emulsion. This work indicated that GNC represented a significantly critical role and properties in the GI tract and reduced lipid digestion and absorption. This GNC could be utilized as an alternative food additive or supplement in fatty food for weight control due to their inhibition of lipid digestibility and assimilation.

## 1. Introduction

Obesity is a severe health problem worldwide, with continuously increasing morbidity and mortality of non-communicable chronic diseases caused by multiple factors. One of the factors is that excessive fatty food consumption leads to the accumulation of excess body fat and suffering from being overweight. The direct uptake of dietary fat in the digestion system occurs as pancreatic lipase is secreted into the small intestine and hydrolyzes triglycerides into monoglycerides and free fatty acids prior to absorption in the small intestine [1]. The primary approach for preventing and treating obesity has been lifestyle modification, including diet and exercise. Unfortunately, lifestyle modifications are often unsuccessful [2]. Consequently, medications have been developed to address overweight and obesity, with the FDA approving five drugs—namely, orlistat (Xenical, Alli), phentermine-topiramate (Qsymia), naltrexone-bupropion (Contrave), liraglutide (Saxenda), and semaglutide (Wegovy) for long-term use. However, these medications are associated with side effects such as diarrhea, stomach pain, dizziness, headache, nausea, vomiting, and increased heart rate. As a result, researchers have been exploring alternative anti-obesity agents derived from plants or other natural sources, including dietary fibers like cellulose [3], pectin [4], the fiber found in pear fruit pomace [5], and cereal [6]. These alternatives have shown potential benefits in controlling obesity, offering a safe, cost-effective, and holistic approach to long-term health promotion. These natural substances have demonstrated the ability to modify the absorption of nutrients and chemicals in the gastrointestinal tract [7], such as by promoting satiation and improving satiety, and preventing obesity [8], as well as delaying digestion and inhibiting enzyme activity [9], this is especially true of nanomaterials like nanocellulose.

Nanomaterials have been used in the human gastrointestinal tract (GIT) as additive and functional foods [10]. In particular, nanocellulose is a cellulose material with a broad spectrum of nanoscale range-based particles with different shapes, sizes, and surface properties [11]. There are many attractive properties of nanocellulose for food applications, such as low cost, biodegradability, renewable nanomaterials, high absorbance, and easy processing. However, it is essential to understanding the behavior, toxicity, and effects on food digestion of nanocellulose for use as a digestion modifier and to limit edible fat absorption. To generate the emulsions, nanocellulose has the ability to form stable oil-in-water (O/W) emulsions through the “Pickering mechanism” in which it forms a protective steric barrier around the oil droplets [12]. Previous studies have suggested that the adsorption of nanocellulose onto lipid droplet surfaces can develop a physical barrier that inhibits the adsorption of lipase and bile salts and retards lipid digestion in GIT. Furthermore, the presence of nanocellulose or dietary fiber in the aqueous phase of emulsions can lead to interactions with bile salts, phospholipids, or calcium, and potentially increase viscosity. These effects can potentially alter lipid digestion [13,14]. Hence, nanocellulose, an insoluble fiber found in nature, holds promise in addressing the potential benefits of controlling lipid digestibility and assimilation.

Based on a previous report, nanocellulose is non-toxic for humans, compatible with biological tissue [15], and cannot be digested in the human GIT. Li et al. [16] investigated three types of nanocellulose, cellulose nanocrystals from a cellulose I crystal structure (CNCs-I), cellulose nanocrystals from a cellulose type II crystal structure (CNCs-II), and cellulose nanofibers (CNFs), on lipid in vitro gastrointestinal digestion using corn oil-in-water emulsions. They showed the different performances during each digestion stage to control the reduction of lipid digestion or release of free fatty acids (FFA). According to FFA determination, the degree of lipid digestion was influenced by both the crystalline structure and form of nanocellulose, especially the morphology. Deloid et al. [17] reported the ability of cellulose nanofibrils (CNF) and cellulose nanocrystals (CNC) to reduce the hydrolysis of fatty foods consisting of heavy cream, coconut oil, mayonnaise, and corn oil. Moreover, CNF and CNC at 0.75% (*w*/*w*) had no significant in vitro toxicity [15]. Some previous research used oil-in-water emulsions with various nanocelluloses to reduce triglyceride hydrolysis in fatty foods and FFA release during simulated in vitro digestion [13,18,19]. By the way, the research on the applications of nanocellulose with a granular or spherical shape in lipid digestibility to release free fatty acids has not been reported. Some previous studies have suggested that granular nanocellulose (GNC) exhibits promising applications as a functional material due to its exceptional thermal stability [20] and highly polydisperse nanoparticles [21]. In addition, Ram et al. [22] reported that GNC had been extensively evaluated in various fields, including synthesizing adsorbents for metal ions in wastewater, supercapacitors, carriers for drug delivery, and cellular uptake.

This research aimed to evaluate the feasibility of the granular nanocellulose particles (GNC) from sugarcane bagasse obtained from enzymatic hydrolysis under optimum conditions to reduce lipid digestion. Determination of the role of GNC in terms of concentration and simulation between GNC and olive oil formation was carried out. Moreover, the characteristics of GNC during the in vitro gastrointestinal simulation (particle size distribution, zeta potential, and interfacial between GNC and oil) were investigated. Furthermore, the cell cytotoxicity on HIEC-6 was assayed using the MTT method. The release of FFA on lipid digestibility and permeability through the HIEC-6 intestinal epithelium was evaluated to indicate the feasibility of reducing fat assimilation for their application as a potential and alternative nano-biomaterial for food additives or supplementation in fatty food for weight control, weight loss, and the management of obesity.

## 2. Materials and Methods

### 2.1. Materials

Potassium chloride (KCl), sodium hydrogen carbonate (NaHCO_3_), and calcium chloride (CaCl_2_) were purchased from LOBA Chemie (Mumbai, India). Potassium dihydrogen phosphate (KH_2_PO_4_) and ammonium carbonate (NH4)_2_CO_3_ were provided from QRëC^®^ (Auckland, New Zealand). Magnesium chloride (MgCl_2_), sodium hydroxide (NaOH), and hydrochloric acid (HCl) were obtained from RCI Labscan (Bangkok, Thailand). Sodium chloride (NaCl) was purchased from CARLO ERBA Reagents (Val-de-Reuil, France). α-amylase (A3176, ≥5 units/mg solid, pepsin (P7000, ≥250 units/mg solid), bile salts (B8756, for microbiology), pancreatin (P7545, 8USP), Calcofluor White Stain, and Nile red were from Sigma-Aldrich (St. Louis, MO, USA). Dimethyl sulphoxide (DMSO) was obtained from Bio Basic Inc. (Markham, ON, Canada). Water was purified with a Milli-Q system (Millipore Milli-Q purification system). 3-(4,5-dimethylthiazol-2-yl)-2,5-diphenyltetrazolium bromide (MTT) was provided from Bio Basic Inc. (Markham, Canada). OptiMEM 1 Reduced Serum Medium, HEPES, GlutaMAX, Epidermal Growth Factor (EGF), and fetal bovine serum (FBS) were from Thermo Fisher Scientific, Gibco (Waltham, MA, USA).

### 2.2. Granular Nanocellulose Particle (GNC) Preparation

The granular nanocellulose particles were produced from the alkaline pretreated and bleached sugarcane bagasse cellulose by enzymatic hydrolysis based on Jirathampinyo et al. [23]. Briefly, sugarcane bagasse was treated with 10% (*w*/*v*) sodium hydroxide at a 1:20 (*w*/*v*) ratio under an autoclave condition for 15 min. The solid residue was washed and dried. Then, the dried solid was bleached by sodium chlorite (2%, *w*/*v*) at 75 °C for 120 min and washed with distilled water to pH 7.0. The extracted cellulose of sugarcane bagasse was obtained and hydrolyzed by enzymatic hydrolysis under optimum conditions. The mixture of extracted cellulose (0.13%, *w*/*v*) and commercial cellulase (containing endoglucanase 174 U/mL) in sodium acetate buffer (0.05 M, pH 5.0) was incubated at 29.5 °C in a shaking incubator with a shaking rate of 120 rpm for 1 h. The mixture was centrifuged at 10,000 rpm for 15 min. The solid residue was dispersed in deionized (DI) water in an ultrasonic bath (The Branson 2510, 40 Hz) for 10 min and centrifuged at 3500 rpm for 10 min. The supernatant was collected and referred to as the granular nanocellulose particle (GNC-E) suspension.

### 2.3. Simulated Gastrointestinal Tract (GIT) Fluids and Enzyme Solutions Preparation

The simulated GIT system and saliva, gastric, and intestinal fluids were prepared. Simulated saliva fluid was adjusted to obtain the final electrolyte concentrations of: 18.875 mM KCl, 17.0 mM NaHCO_3_, 4.625 mM KH_2_PO_4_, 0.06 mM (NH_4_)_2_CO_3_, and 0.05625 mM MgCl_2_. In the simulated gastric fluid, 59 mM NaCl, 31.25 mM NaHCO_3_, 8.625 mM KCl, 1.125 mM KH_2_PO_4_, 0.625 mM (NH_4_)_2_CO_3_, and 0.15 mM MgCl_2_ were prepared. Then, the pH value of 3 was adjusted with 1 M HCl. Likewise, simulated small intestinal fluid was designed to a final concentration of electrolytes of 106.25 mM NaHCO_3_, 48 mM NaCl, 8.5 mM KCl, 1 mM KH_2_PO_4_, and 0.4125 mM MgCl_2_. The pH value was adjusted to 7 with 1 M HCl. Enzyme solutions were prepared daily in each simulated fluid to receive the concentration of α-amylase (1000 mg/L), pepsin (31,660.61 mg/L), and pancreatin (8000 mg/L). Before use, each individual enzyme solution was pre-incubated at 37 °C. Bile salt was prepared to obtain a concentration of 25,000 mg/L in simulated small intestinal fluid.

### 2.4. In Vitro Simulated GIT Digestion System

All simulated GIT fluids and enzyme solutions were prepared and pre-incubated at 37 °C before being used in the in vitro simulated GIT digestion. The procedure was based on Jakobek et al. [24]. The oral, stomach, and small intestinal stages were consistently in the system throughout the simulated GIT model. The ingested GNC suspension and GNC–olive oil mixture were added at the beginning of the digestion system. The initial solution was added to the oral phase containing 3.5 mL of simulated salivary fluid, 975 μL of H_2_O, 25 μL of CaCl_2_ (0.3 M), and 500 μL of α-amylase and mixed by vortex for 30 s. The oral phase solution was mixed with the simulated gastric fluid (7.5 mL), 295 µL of H_2_O, 5 µL of CaCl_2_ (0.3 M), 200 µL of HCl (1 M), and 2 mL of pepsin. The mixture was vortexed and incubated in a water bath with shaking for 2 h at 37 °C. The simulated intestinal phase contained 11 mL of intestinal fluid, 3.61 mL of H_2_O, 40 µL of CaCl_2_ (0.3 M), 150 µL of NaOH (1 M), 5 mL of pancreatin, and 0.2 mL of bile salt. After that, the simulated oral and gastric digest solution was added and mixed with 150 μL of NaOH (1 M), 5 mL of pancreatin, and 0.2 mL of bile salt. The intestinal phase mixture was incubated in a shaking water bath for 2 h at 37 °C. The entire solution from each stage was collected to analyze the characteristics of GNC during the GI tract system.

### 2.5. The Role of GNC in Releasing FFA Content in the Simulated GIT System

#### 2.5.1. GNC Concentrations

The granular nanocellulose particle concentrations (0.01–0.08% *w*/*v* of the final concentration) were investigated for their effects on releasing free fatty acids in GIT simulation. The suspension was mixed with olive oil by vortex for 30 s, followed by simulated gastrointestinal tract model digestion. The final digesta after simulation were titrated with 50 mM sodium hydroxide using thymolphthalein as an indicator. The released FFA concentration was determined.

#### 2.5.2. The Simulation Mixture of the GNC and Olive Oil

The GNC and olive oil (ratio 3:1, *w*/*w*) simulation mixtures were prepared by vortex mixing for 30 s to obtain the GNC mixing form, and the GNC emulsion form was generated by a high-intensity ultrasonic processor (VCX, USA) using 500 watts at the amplitude of 40%, pulses of 2 s on and 3 s off for 21 s. Both GNC and olive oil simulation mixtures were added to the GIT system. The final digestion products were titrated with 50 mM sodium hydroxide using thymolphthalein as an indicator. The measurement of released free fatty acid content was conducted.

### 2.6. The Characteristics of GNC during In Vitro GIT Simulation

#### 2.6.1. The Particle Size Distribution and Zeta Potential Determination

A Zetasizer (Malvern Nano particle analyzer series, Nano ZS) was used to determine the particle size distribution and zeta potential value of the GNC during in vitro GIT simulation. The reflective index at 1.47 and a temperature of 25 °C with 0.01% of each sample were measured.

#### 2.6.2. The GNC–Olive Oil Droplets Characterization

Simulated GNC and olive oil formation (GNC–olive oil mixture and emulsion) in the initial stage prior to ingestion into the GIT simulation was observed by bright-field microscopy (CX43, Olympus corporation Japan) using 20× objective lenses. The oil droplet was stained with Nile red.

#### 2.6.3. The Structural Characterization of the GNC–Olive Oil Emulsion

The GNC–olive oil emulsion at the initial and final digesta stages of GIT simulation was detected by a confocal laser scanning microscope (ZEISS LSM 900) using 60× objective lenses. The oil droplet and GNC particles were stained with Nile red and calcofluor white, respectively.

### 2.7. Cytotoxicity of GNC

HIEC-6 (ATCC^®^ CRL-3266^TM^) cells were obtained from the American Type Culture Collection (ATCC) and were cultured in the completed medium using an OptiMEM 1 reduced serum medium containing 20 mM HEPES, 10 mM GlutaMAX, 10 ng/mL epidermal growth factor (EGF), and 4% fetal bovine serum (FBS) and incubated at 37 °C in a humidified atmosphere of 95% air and 5% CO_2_. The HIEC-6 cultures were seeded and maintained in a 96-well plate at a 20,000 cells/well density until 90% confluence. After the 24 h preconditioning, cell cultures were treated with various concentrations of GNC (0.01–0.50% *w*/*v*) and incubated at 37 °C in a humidified atmosphere of 5% CO_2_ for 24 h using the medium as the control. The 3-(4,5-dimethylthiazol-2-yl)-2,5-diphenyltetrazolium bromide (MTT) assay [25] was used for determining cell viability. The MTT solution (concentration 5 mg/mL in phosphate buffer saline; PBS) was added and then the solution continued to be incubated for 4 h. After removing the MTT solution, DMSO was added to dissolve the formazan crystals to determine the absorbance at 550 and 620 nm using a microplate reader (VarioskanTM Flash Multimode Reader, Thermo scientific^®^, Waltham, MA, USA). Compared to the control, the percentage of cell viability was estimated.

### 2.8. The Lipid Digestibility and Permeability of the HIEC-6 Cell Monolayer

The final digesta after simulation was treated in the intestinal phase for 2 h digestion times. The FFA concentration from lipid digestibility was determined using a colorimetric assay kit. Figure 1 demonstrates a schematic of a transwell support system for a permeability assay of HIEC-6 cell monolayer. HIEC-6 cells were seeded at 1 × 10^5^ cells/insert on polyethylene terephthalate (PET) micropore (0.4 µm of diameter) membranes into transwell inserts placed in a multiwell 24 plate and then incubated at 37 °C in a humidified atmosphere of 5% CO_2_. The medium was changed every two days until day 8, and the transepithelial electrical resistance (TEER) was measured using a volt-ohmmeter to assess the integrity of the HIEC-6 monolayer. The final digest solutions of the GIT digestion were mixed with the medium in a 1:3 (*v*/*v*) ratio and treated on the apical side. After adding the digestion solution for 2 h, the free fatty acid and total triglyceride contents of the basolateral sides were determined using a colorimetric assay kit.

### 2.9. Statistic Analysis

The experiments were carried out in triplicate, and their statistical significance was expressed in terms of average mean and standard deviations (SD). IBM SPSS Statistics Windows version 25.0 (IBM Corp, Armonk, NY, USA) was used to evaluate the statistical differences using one-way ANOVA at a *p*-value of less than 0.05 (*p* < 0.05).

## 3. Results

### 3.1. The Characteristics of Granular Nanocellulose Particles (GNC) from Sugarcane Bagasse

The morphology of the produced GNC obtained from sugarcane bagasse by enzymatic hydrolysis was observed by a field emission scanning electron microscope and is demonstrated in the uniform and aggregated particles, as shown in Figure 2. The diffraction intensity of GNC was analyzed by an X-ray diffractometer and revealed a slight decrease in the crystallinity, with a Crl value of about 30.6 compared to the original cellulose. The particle size distributions of GNC were analyzed by a Zetasizer. The size of GNC particles was in the range of 220–458 nm in diameter, with an average of 342 nm, and the zeta potential was −49.0 mV, leading to their determination as suitable for good suspension stability with a negative charge. Moreover, the polydispersity index (PDI) of the produced GNC was 0.431, indicating the considerable homogeneity of the GNC size.

### 3.2. The Role of GNC in Releasing FFA Content after In Vitro Gastrointestinal Digestion

The influence of GNC on triacylglycerol hydrolysis in the gastrointestinal tract was investigated. Alkali titration determined the released FFA content using olive oil as the triacylglycerol source. GNC concentrations and GNC–olive oil simulation mixtures were conducted to evaluate the effect of releasing free fatty acids after in vitro gastrointestinal digestion.

#### 3.2.1. The Role of GNC Concentration

The effect of GNC concentration on releasing FFA from olive oil hydrolysis in the simulated gastrointestinal tract was evaluated and is shown in Figure 3. The results indicated that the GNC concentration of 0.02–0.08% (*w*/*v*) significantly affected the liberation of FFA, resulting in a decrease in FFA concentration compared to without GNC. At 0.08% GNC concentration, the highest decreasing FFA content was shown and the FFA concentration of about 138.3 ± 3.4 µM was found, with a 20% reduction compared to the control (180.0 ± 6.5 µM). However, there were no significant differences in releasing FFA when using a 0.02–0.08% (*w*/*v*) GNC concentration on olive oil digestion under simulated intestinal conditions.

#### 3.2.2. The Role of the Simulation Mixture of GNC and Olive Oil

The different simulation mixtures of GNC and olive oil were prepared as shown in Figure 4a,b to investigate their influence on releasing free fatty acids in the in vitro gastrointestinal tract system. The GNC–olive oil emulsion (Figure 4b) displayed homogeneity in the GNC–olive oil phase. In the GNC–olive oil mixture presented inversely, the coalescence of oil droplets occurred, resulting in the separation of the oil and GNC suspension phase (Figure 4a). The results explained that the stability of the GNC–olive oil mixture was not good enough. To understand the interfacial phenomenon, the oil droplet size and distribution in the GNC–olive oil mixture and the Pickering emulsion were estimated, and the optical microscopy photographs are illustrated in Figure 4c and Figure 4d, respectively. The oil droplets in the GNC–olive oil mixture prepared by vortex mixing revealed large droplet sizes and a heterogeneous distribution (Figure 4c). The microscopy photography observed droplets aggregated into clusters in the oil-to-GNC suspension mixture. For the Pickering emulsion of GNC–olive oil, the small oil droplets were suspended within the emulsion, as shown in Figure 4d. Furthermore, the behavior of GNC–olive oil emulsion during simulated gastrointestinal digestion significantly reduced the olive oil hydrolysis, with approximately 22% hydrolysis inhibition (Figure 4e).

### 3.3. Characteristics of GNC in the In Vitro Simulated Gastrointestinal System

#### 3.3.1. The Particle Size Distribution and Zeta Potential Value

The particle size distributions of the ingested GNC suspension compared to the GNC–olive oil emulsion in the simulated GI tract are shown in Figure 5. In addition, Table 1 summarizes the average particle size, the polydispersity index (PDI), and the zeta potential value of each suspension. In the initial stage, the particle size distributions of the ingested GNC suspension (Figure 5a) and GNC–olive oil emulsion (Figure 5b) were similar in diameter, between 190 and 458 nm, with an average size distribution of 295.3 nm. The polydispersity indexes (PDI) of both the GNC suspension and GNC emulsion were 0.551, presenting monodispersity. Moreover, the zeta potential remained within −38.2 mV, indicating good stability of the GNC suspension and GNC emulsion.

In the oral phase, the ingested GNC suspension and GNC–olive oil emulsion showed different size distributions, as shown in Figure 5a and Figure 5b, respectively. The results revealed that of the ingested GNC particles, approximately 90% were distributed in the 342–615 nm range, slightly higher than in the initial stage due to GNC particle swelling and fluctuation in the oral phase. Therefore, the PDI value was 0.835, indicating a broad particle size distribution but still a good dispersity. The GNC emulsion in the oral phase demonstrated that the particle size of the GNC (72.5% of the total particle) was significantly increased and varied between 1484 and 2669 nm, with about a 20% intensity of particle size at 1990 nm, and the zeta value was −7.56 mV. However, the particle size distribution presented was homogenous, with a PDI value of 0.597, due to the formed emulsion with olive oil leading to the GNC–oil droplet particles increasing in particle size.

The particle size distributions of the ingested GNC suspension significantly increased during the stomach phase. Figure 5a,b illustrates the GNC particle size of around 825 nm with 55% intensity, respectively. In contrast, the size of GNC in the emulsion decreased into a range of 1106–1484 nm, with an average size of 1281 nm, compared to the oral phase. The results indicated that both ingested GNC and GNC–olive oil emulsions have a high homogenous particle size during the gastric phase, with PDI values of about 0.111 and 0.352, respectively. The zeta potential value of the GNCs in the stomach phase was −10.5 mV for ingested GNC and −0.5 mV for the GNC–olive oil emulsion.

During small intestinal digestion, the GNC particles, approximately 77% of them, returned to the initial size in the 342–458 nm range of the ingested GNC suspension, with a broad distribution and less stability. In contrast, all GNC particle sizes in the emulsion were inverse to 220 nm with a zeta potential of −33.3 and had excellent homogeneity and stability (PDI = 0.111).

#### 3.3.2. The Structural Characterization of GNC–Olive Oil Emulsion in the Simulated GIT System

Visual observations via confocal laser scanning microscope imaging revealed that the GNC interfaced with olive oil to generate the emulsion droplet during the digestion stage of the GIT, as shown in Figure 6. The GNC particles are represented by the blue color of the Calcofluor white staining, while oil droplets, shown in red, were stained with Nile red. The GNC–oil droplets were dispersed in the oil-to-water (Pickering) emulsion in the initial phase. Figure 6a demonstrates that the GNC particles distributed surrounding the olive oil droplets and obviously interacted with the oil molecule, as shown in the zoomed image in Figure 6b. After the intestinal phase digestion of the GIT simulation, the oil droplets were revealed to be small in size and there were less than in the initial stage, primarily due to lipid digestion resulting in the free oil droplet distribution (Figure 6c).

### 3.4. Cytotoxicity of GNC during the GIT Simulation

The cytotoxicity of GNC to HIEC-6 (ATCC^®^ CRL-3266^™^) cells was investigated, and the cell viability was evaluated using the MTT assay. The results revealed that GNC had low cytotoxicity at tested concentrations (0.01–0.10% *w*/*v*) and demonstrated high cell viability, with more than 80% cell viability, ranging between 80.0 ± 4.1 and 99.0 ± 3.3% compared to that without GNC (Figure 7). However, a higher GNC concentration at 0.50% (*w*/*v*) found a cell toxicity effect with 68.3 ± 7.2% cell viability. The results were consistent with previous studies investigating the impact of nanocellulose materials, including cellulose nanocrystals and cellulose nanofibrils, on in vitro toxicity in gastrointestinal (GIT) cells.

### 3.5. The Release of FFA in the Simulated GIT System

The influence of GNC on lipid digestion during a simulated GIT system was investigated by estimating the released FFA content at 0 and 2 h of digestion times. The results compared the lipid digestibility in digesta with and without ingested GNC suspension and GNC–olive oil emulsion. The concentration of released FFA increased when digestion time increased. As shown in Figure 8, the increase in released FFA occurred in those without GNC digesta more than in those containing GNC after a digestion time of up to 2 h. The FFA concentration increased by approximately 90% without GNC, while it increased by about 61.2% and 69.9% in the digesta containing ingested GNC suspension (Figure 8a) and GNC–olive oil emulsion (Figure 8b). Moreover, the FFA contents in the digesta containing ingested GNC suspension and GNC emulsion were lower than those without GNC, with a reduction of about 30% in lipid digestion. The results indicated that GNC affected the reduction of lipid digestibility, leading to a decreased FFA being released during the gastrointestinal simulation.

### 3.6. The FFA and TG Permeability in HIEC-6 Cell Monolayer

The integrity of the permeability was evaluated by the transepithelial electrical resistance (TEER) value of the HIEC-6 cell monolayer growing on the transwell insert membranes. The TEER value of the functional HIEC-6 cell monolayer was 108 ± 12 ohm·cm^2^ on day 8, indicating that the monolayers remained intact.

The results revealed the permeability of the final digesta products from the GI tract simulation containing the ingested GNC suspension and GNC–olive oil emulsion compared to that without GNC. The FFA and triglyceride (TG) concentrations were detected on the basolateral side after 2 h of permeable time. The results showed that the FFA contents significantly decreased with the ingested GNC suspension (Figure 9a) and GNC–olive oil emulsion (Figure 9b) compared to without GNC. These results indicated that GNC reduced fatty acid permeability across the HIEC-6 monolayer. In contrast, the results presented no significant difference in TG contents on the basolateral side with and without GNC, resulting in GNC not affecting the generation of TG after permeation through the HIEC-6 intestinal epithelium (Figure 9c,d). These results suggested that the ingested GNC and GNC–olive oil emulsion affected the reduction of FFA permeability of the HIEC-6 monolayer.

## 4. Discussion

The granular nanocellulose particles (GNC) produced from sugarcane bagasse by enzymatic hydrolysis were investigated for their role in releasing free fatty acids (FFA) in the simulated GIT system. The results presented that a 0.02–0.08% (*w*/*v*) GNC concentration significantly decreased the release of FFA in olive oil hydrolysis under simulated intestinal conditions. The effect of GNC in releasing FFA content after in vitro gastrointestinal digestion was consistent with the work of Liu and Kong [26], who reported that nano-fibrillated cellulose of 0.22 and 1.1% (*w*/*w*) had no significant differences in decreasing the amount of FFA released at the end of intestinal digestion. In addition, high concentrations of 1% cellulose nanofiber (CNF), 0.25–0.36% TEMPO-CNF, and 2–3% cellulose nanocrystals (CNC) [27] revealed a delayed initial release of free fatty acids during the digestion of Tween 80-stabilized lipid emulsions. However, the differences in fiber source, purity, dosage, experimental conditions, and test methodologies have contributed to the different results reported by other studies [26]. Previous research has suggested that the increment in nanocellulose content can increase viscosity, which may impact lipase activity [28]. Viscous fibers have been found to reduce triacylglycerol hydrolysis, thereby decreasing the area available for lipase access [29] and lowering the release of FFA content. Additionally, the polar or ionic fiber site may interact with lipase, leading to a reduction or inhibition in pancreatic lipase activity, according to studies conducted by Chen et al. [30], Skjold-Jørgensen et al. [31], and Yu et al. [32].

Moreover, the simulation mixture of GNC and olive oil was prepared in a vortexed mixture and the emulsion was used to investigate the influence on reducing FFA. The results showed that the GNC–olive oil emulsion reduced olive oil hydrolysis, with appropriately 22% inhibition. The role of the simulation mixture of the GNC–olive oil emulsion displayed the homogeneity of the GNC–olive oil phase, indicating the Pickering emulsion formation of nanocellulose-stabilized products [33]. The results demonstrated that the GNC colloidal particles enhanced the emulsion stability and protected their aggregation by absorbing the oil droplet surfaces. The results corresponded to Wen et al. [34], who reported that larger emulsion droplets were much less stable under low shear by mixing than smaller droplets formed at higher nanoparticle concentrations. Moreover, nanocellulose satisfies the increasing demands for a sustainable and environmentally friendly stabilizer. Nanocelluloses are likely to form o/w emulsions, which are an emulsion stabilized by solid particles (Pickering emulsions) [35] due to their amphiphilic surface nature, which originates from the hydrophobic face and hydrophilic edge of cellulose chains [36]. This formation offers a wide range of potential applications such as drug delivery, food, and composite materials because it generally provides a more stable system than surfactant-stabilized emulsions.

The particle size and distribution, the zeta potential, and PDI values were measured to understand the behavior and the important role of GNC in the GI tract. Moreover, the simulation mixture of GNC and olive oil and Pickering emulsion was observed to determine the morphological stability during all digestion phases. According to the characteristics of GNC in the initial phase, an average size distribution of 295.3 nm with a zeta potential of −38.2 mV and PDI value of 0.551 was represented. A zeta potential value of around 30 mV with a negative charge is suitable for maintaining the stability of the suspension and preventing the nanocellulose from aggregating [37]. In the oral phase, the results revealed that the GNC particles were swelling and fluctuating, with an average size at 1990 nm, and the zeta value was −7.56 mV, resulting in a low dispersion of the GNC emulsion, causing the aggregation of GNC. The results concurred with Capron et al. [38] who found that various nanocellulose types, including cellulose nanocrystals and cellulose nanofibers with different sizes and lengths, maintained their particle size stability during the oral stage. However, some studies have suggested that smaller cellulose nanocrystals lead to flocculation during the oral phase [39] and swelling after entering the oral stage. This effect may result from anionic mucin molecules in the simulated saliva fluid adsorbing to the surfaces of particles [40]. In the gastric phase, the ingested GNC suspension and GNC emulsion were unstable due to the zeta potential values being lower than −2.00 mV because of the low pH value and high ionic strength. These factors compressed the electric double layer, leading to a decrease in the zeta potential value [16,41,42]. The increasing size of GNC indicates the occurrence of increased swelling and flocculation of nanocellulose particles, which is consistent with prior research [41,43,44]. Moreover, the gel-like structure resulting from the aggregation of oil droplets in the gastric environment was caused by the negative charge of nanocellulose, which is highly susceptible to alterations in ionic strength [39,45]. In the small intestinal phase, the zeta potential values experienced a significant increase due to the neutral pH conditions. The increase in pH values could help eliminate the charge shielding of nanocellulose. Additionally, the absorption of many anions, such as bile salts, free fatty acids, and phospholipids, onto the lipid surface can enhance the negative surface charge [17]. This increase in particle size can be attributed to the highly negative zeta potential increasing the electrostatic repulsive force among lipid droplets [41], as well as various relatively small colloidal particles, including micelles, vesicles, and insoluble calcium soaps, possibly forming during the lipid digestion process [46].

These results noted that the presence of food, such as olive oil, and various parameters in the gastrointestinal tract, including residence time, pH, ion strength, mechanical force, and GIT secretions, such as enzymes, bile salts, acids, and hormones have effects on behavior [47]. Likewise, the presence of a food matrix can also affect the behavior of nanocellulose in the gastrointestinal tract [19,26]. In GIT simulation, the stabilized lipid emulsion becomes unstable, and the surfactants adsorb more strongly to the droplet surfaces, resulting in the large surface area of lipids exposed to lipase and gastrointestinal fluids, thereby presenting a low efficiency at inhibiting lipid digestion [48]. Mun et al. [49] reported that the stability of lipid droplets against disruption and coalescence during lipid digestion is primarily determined by the hydrophilic and hydrophobic properties of the emulsifiers coating the interfacial layer of the droplets. Additionally, the properties of this layer can impact the hydrolysis of lipid droplets by lipases in the small intestine. Nanocellulose, which can stabilize emulsions, has numerous potential uses due to its amphiphilic surface characteristics. This property makes it a more reliable stabilizer than surfactants, as it arises from the hydrophilic edge and hydrophobic face of cellulose chains [35]. The structural characteristic of GNC–olive oil emulsion droplets remained in the digesta of the intestinal phase after the simulation of GIT digestion, as can be seen in the zoomed in image in Figure 6d. Hence, it can be inferred that GNC potentially alters the colloidal interactions among the oil droplets, affecting their aggregation state and the surface area of lipids that is accessible to enzyme activity. This hypothesis is consistent with previous research findings [17,39,50].

However, the GNC has yet to be accepted as safe or approved for use as a food ingredient or to be ingested separately. Thus, investigation of the toxicology of ingested GNC is of considerable importance. Study of the cytotoxicity of GNC in the GIT simulation showed less toxicity, which is supported by Vital et al. [51], who showed that the ingested nanocellulose is not toxic to the gastrointestinal tract. Furthermore, the non-toxicity of nanocellulose, focusing on natural sources, was observed at concentrations ranging from ~0.2% to 1.0% (*w*/*w*) [52]. Deloid et al. [15] reported that the ingested nanocellulose had slightly acute toxicity and was not a risk when consumed in small amounts. The results found that cellulose nanofibrils (50 nm width) and cellulose nanocrystals (25 nm width) at 0.75% (*w*/*w*) did not exhibit any toxicity in the in vitro studies.

Furthermore, the effect of GNC on lipid digestibility and permeability was evaluated for the feasibility of using GNC to improve food quality, especially high-fat food, and for use as a supplement for weight control. The results revealed that the FFA content in the digesta containing an ingested GNC suspension and GNC emulsion was lower than without GNC and resulted in a decrease of 30% in lipid digestion. The influence of GNC on lipid digestion during a simulated GIT system corresponded to Rungraung et al. [40], who indicated that nanofibrillated cellulose (NFC) or nanocrystalline cellulose (NCC) at a concentration range between 0.05 and 0.20% *w*/*w* on oil-filled beads decreased the rate of lipid digestion; 0.20% (*w*/*w*) NFC and NCC showed an FFA content of the final digesta of around 60% and 49%, respectively. Ni et al. [39] showed that the effects of cellulose nanoparticles on the gastrointestinal fate and emulsion digestion (10% oil *v*/*v*) reduced the FFA released by 71.4 ± 1.5% and 49.6 ± 2.1% when using cellulose nanoparticle sized at 310 ± 15.9 nm and 955 ± 20.6 nm, respectively. In addition, Liu and coworkers [27] studied the three types of nanocellulose, including cellulose nanocrystal (CNC), cellulose nanofibril (CNF), and TEMPO-oxidized cellulose nanofibrils (TEMPO-CNF) on lipolysis rate with released FFA content. The results demonstrated that 1% CNF, 0.25–0.36% TEMPO-CNF, or 2–3% CNC delayed initial in vitro digestion of emulsions, though the final lipolysis extent was nearly the same amongst all (47–55%).

The influence of an ingested GNC suspension and GNC emulsion on FFA and TG permeability across the HIEC-6 cell monolayer was also investigated. The morphology features of the HIEC-6 cell monolayer, which studied the FFA and TG permeability, were similar to the human small intestine [53], whereas the most frequently applied cellular model was the human colon carcinoma Caco-2 cell line [51]. According to Takenaka [54], HIECs showed permeable values of paracellularly absorbed hydrophilic molecules, and their TEER value on day 5 was 98.9 ± 17.5 ohm·cm^2^, which remained constant on day 8. This value was similar to the TEER values reported for the human small intestine (duodenum, jejunum, and ileum). The permeability of the final digesta products (FFA and TG) revealed that the digesta containing the GNC suspension and GNC emulsion reduced FFA permeability, while no significant difference in TG content through the HIEC-6 intestinal epithelium was observed. These results corresponded to DeLoid et al. [17], who discovered that the presence of cellulose nanofibrils (CNF) resulted in a decrease of approximately 52% and 32% in FFA and TG concentration, respectively, on the basolateral side after a 2 h absorption period. Furthermore, Liu et al. [27] demonstrated that 1% CNF or 2–3% CNC delayed the digestion of lipid emulsion by approximately 47–55%. Bai et al. [55] and Winuprasith et al. [13] reported that CNF and CNC decreased lipid digestion and FFA content. The results suggested that the slower rate of lipid digestion was possibly due to three mechanisms. Firstly, the coalescence of oil on nanocellulose reduced the surface area for lipase binding. Secondly, the sequestration of bile salts by nanocellulose impaired interfacial displacement, and solubilization of lipid digestion products by bile salts occurred [17,39]. Thirdly, nanocellulose may alter the colloidal interactions between the lipid droplets, which changes their aggregation state and the surface area of lipids exposed to lipase [50].

## 5. Conclusions

This research indicated the critical role of granular nanocellulose particles (GNC) from sugarcane bagasse in reducing lipid digestibility and absorption in the in vitro simulated gastrointestinal tract (GIT) system. GNC concentrations (0.02–0.08%, *w*/*v*) significantly affected free fatty acid (FFA) release. The simulation formulation model between GNC and olive oil as the Pickering emulsion revealed increased oil droplet size distribution and stability in the initial stage before adding it into the GIT system. The particle size distribution and zeta potential value of the ingested GNC suspension and GNC–olive oil emulsion were different in size distribution and dispersity during the in vitro GIT simulation. The interface of GNC and oil droplets was investigated by confocal laser scanning microscopy. The cell cytotoxicity on HIEC-6 demonstrated low toxicity at the tested concentration (0.01–0.10%, *w*/*v*) with 80% cell viability. The release of FFA on lipid digestibility and permeability through the HIEC-6 intestinal epithelium monolayer was decreased in the digesta containing the ingested GNC suspension and GNC–olive oil emulsion. This work suggested the role of GNC in reducing lipid digestion and absorption during in vitro GIT simulation. GNC will be further used as an alternative nanomaterial for food additives or supplements in fatty food for weight control due to their inhibition of lipid digestibility and reduction of lipid absorption.

## Figures and Tables

**Figure 1 biomolecules-13-01479-f001:**
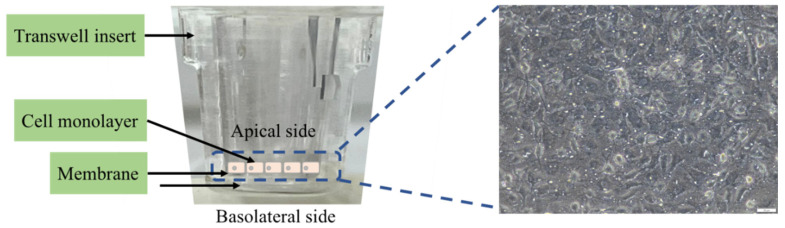
Schematic of a transwell support system for permeability assay of the HIEC-6 cell monolayer and morphology of the HIEC-6 cell line (day 8), 20×.

**Figure 2 biomolecules-13-01479-f002:**
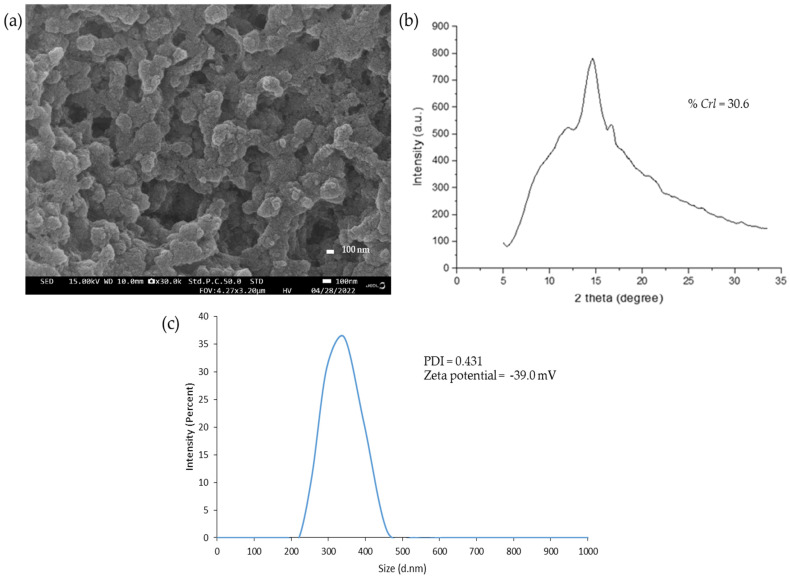
The characteristics of the produced GNC from sugarcane bagasse by enzymatic hydrolysis. (**a**) SEM images (30,000×), (**b**) the XRD pattern with the *%Crl*, (**c**) the particle size distribution.

**Figure 3 biomolecules-13-01479-f003:**
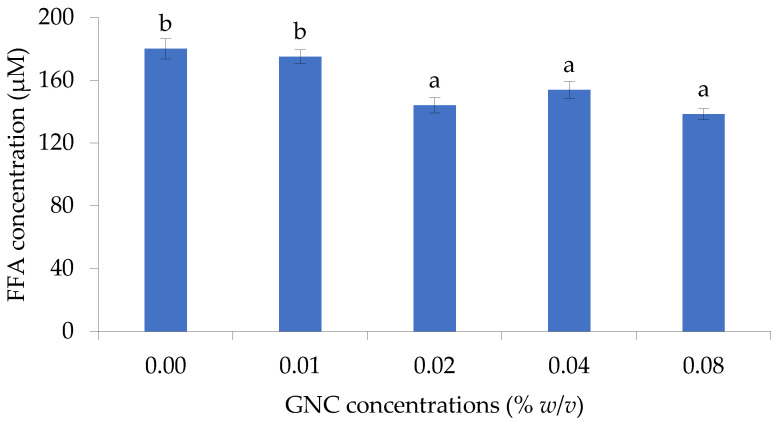
The release of FFA from olive oil digestion in the gastrointestinal tract (GIT) simulation under different GNC concentrations (Different lowcase letters indicated statistically significant difference, *p* ≤ 0.05).

**Figure 4 biomolecules-13-01479-f004:**
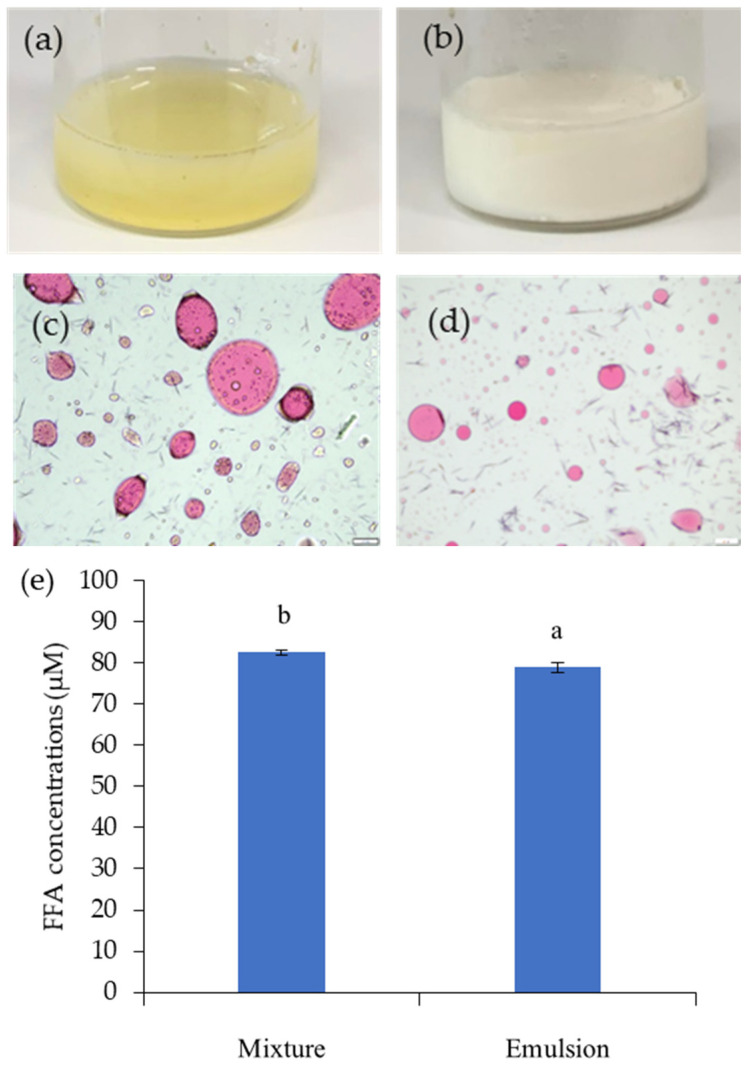
The digital images of simulation mixtures of GNC and olive oil. (**a**) GNC–olive oil mixture and (**b**) GNC–olive oil emulsion; the optical microscopy images of (**c**) GNC–olive oil mixture and (**d**) GNC–olive oil emulsion and (**e**) the free fatty acid concentration after olive oil hydrolysis in the small intestinal stage of the GIT system (Different lowcase letters indicated statistically significant difference, *p* ≤ 0.05).

**Figure 5 biomolecules-13-01479-f005:**
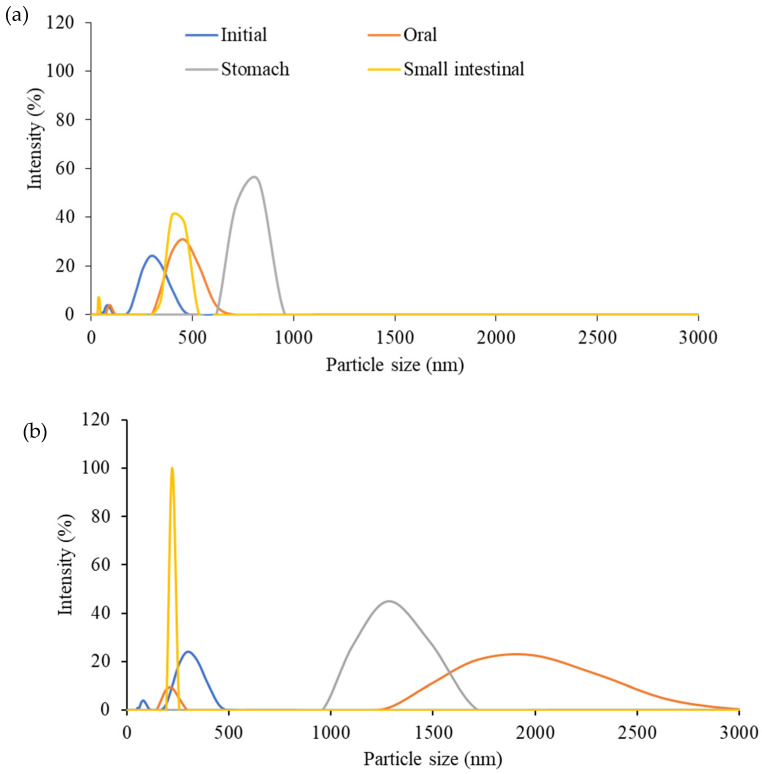
The particle size distributions of (**a**) ingested GNC suspension and (**b**) GNC–olive oil emulsion during in vitro gastrointestinal tract at different GI stages.

**Figure 6 biomolecules-13-01479-f006:**
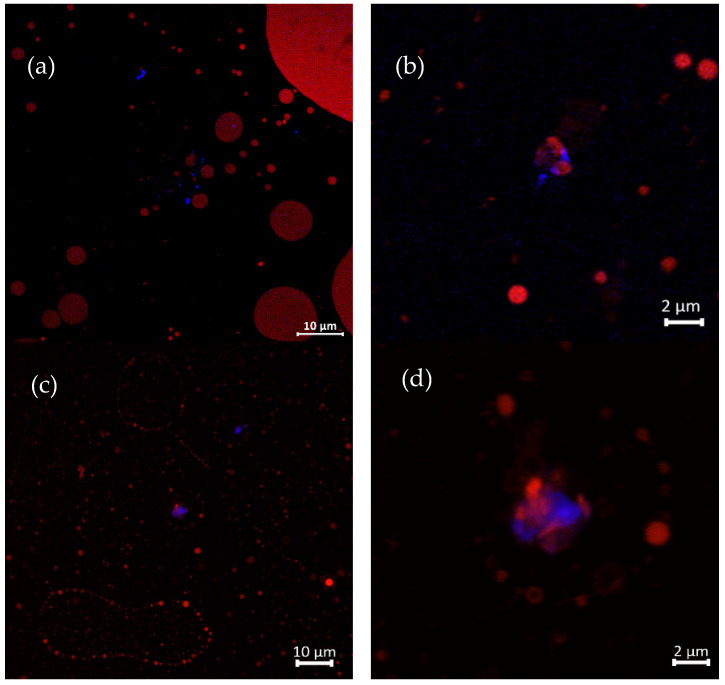
Confocal laser scanning microscopic images of GNC–olive oil interaction (**a**) in the initial stage, 60× (**b**) initial stage, zoomed 300×, and (**c**) in the digesta after GIT simulation, 60×; (**d**) in the digesta after GIT simulation, zoomed 300×. Noted: GNC particles (stained in blue) and olive oil (stained in red).

**Figure 7 biomolecules-13-01479-f007:**
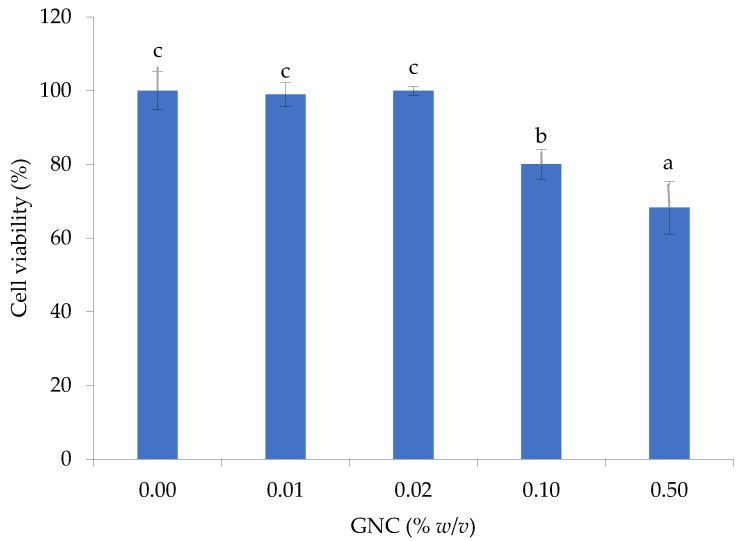
The toxicity on HIEC-6 cell viability of GNC at various concentrations after 24 h (Different lowcase letters indicated statistically significant difference, *p* ≤ 0.05).

**Figure 8 biomolecules-13-01479-f008:**
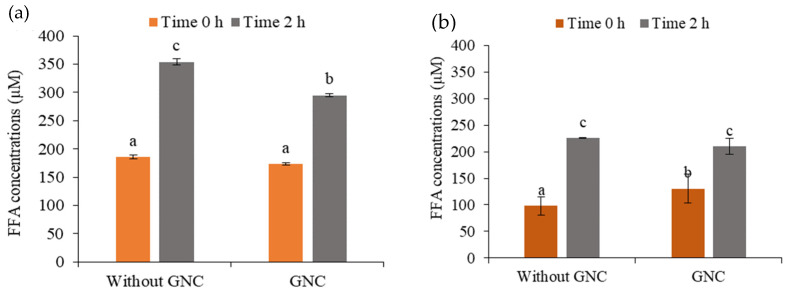
The release of free fatty acids in the intestinal digestion solution at 0 and 2 h of digestion time containing (**a**) the ingested GNC suspension and (**b**) the GNC–olive oil emulsion (Different lowcase letters indicated statistically significant difference, *p* ≤ 0.05).

**Figure 9 biomolecules-13-01479-f009:**
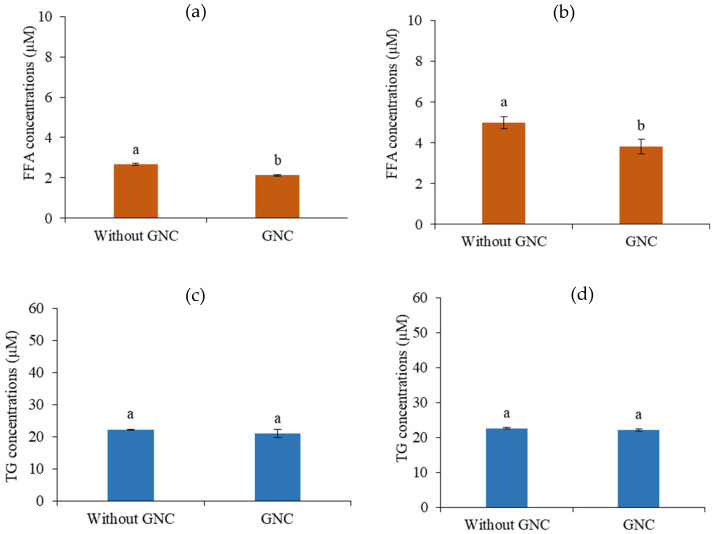
The fatty acid concentration in the basolateral phase of (**a**) the ingested GNC and (**b**) the GNC–olive oil emulsion, and the triacylglycerol permeability (absorption) of (**c**) the ingested GNC and (**d**) the GNC–olive oil emulsion through HIEC-6 monolayer (Different lowcase letters indicated statistically significant difference, *p* ≤ 0.05).

**Table 1 biomolecules-13-01479-t001:** The particle size distributions at the highest intensity, polydispersity indexes (PDI), and zeta potential values of the ingested GNC suspension and GNC–olive oil emulsion in the in vitro GI tract.

Sample	Characteristics	Initial Phase	Oral Phase	Stomach Phase	Small Intestine Phase
Ingested GNC suspension	Particle size (nm)	342.0	458.7	825.0	458.7
	PDI	0.431	0.835	0.211	0.826
	Zeta potential (mV)	−49.0	−33.8	−10.5	−31.4
GNC-olive oil emulsion	Particle size (nm)	295.3	1990.0	1281.0	220
	PDI	0.551	0.597	0.452	0.111
	Zeta potential (mV)	−38.2	−7.56	−0.5	−33.3

## Data Availability

The research created experimental data that can be found in the tables and figures presented in this manuscript.

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
