# Peer review of "Important Role and Properties of Granular Nanocellulose Particles in an In Vitro Simulated Gastrointestinal System and in Lipid Digestibility and Permeability"

_biomolecules, 2023, doi:10.3390/biom13101479_

Round 1

Reviewer 1 Report

Dear Authors;

Thank you very much for this excellent article. Here some comments and suggestions.

The research evaluated the role and feasibility of granular nanocellulose particles in reducing lipid digestibility and permeability in in-vitro simulated gastrointestinal system. 

1) It is suggested to include a brief explanation of why and which information is pretended to obtain from the experimental procedures described in sections 2.6, 2.7 and 2.8. 

2) It is suggested to separate Results and Discussion in two sections. During the reading it was hard to follow the 8-9 tests and their corresponding interpretation. Moreover, some times the own results are not clearly differentiated from the results extracted from the literature used to reinforce the validity of the discussion.

Reviewer 2 Report

In present study, the impact of GNC from sugarcane bagasse on the in vitro digestion properties and cell cytotoxicity was investigated to explore their potential application as novel food additive for weight control and obesity management. The research point was interesting. However, there were some queries about the research methods and results, as shown in belows.

Q1: Please keep a uniform format through the manuscript. Line 127, 136, 146,213,210...delete the extra space to make the paper more readable.

Q2:Line 131-132 Whats the centrifugation speed for the mixture of celluloas and cellulose?

Q3:Line175-176, for the preparation of GNC emulsion, what was the sonication power? Please provide the detailed information.

Q4:Line 190, the title should be revised

Q5:Line 234-242, AFM or TEM technique should be more suitable for observing the structure or morphology of GNC. The observation by SEM was not clear. How the crystallinity, particle size distribution was determined in present study since the relative results were not provided in the Figures.

Q6:Line 181-184, the Pickering emulsion droplets formed by nanocellulose were of micro grade. Is it feasible to measure the particle size with Malvern Nano particle analyzer series (Nano ZS). As we could find the result from Fig.5, the particle size of Pickering emulsion even reached 2.5 μm. I suggest the microscopic observation of Pickering emulsions and using the ImageJ tool to measure the particle size.

Q7:In Table1, why the particle size, PDI and zeta-potential of ingested GNC suspension and GNC emulsion were the same ?

Q8: In Figure 6(a), the GNC which indicated by blue color should distribute around the droplets. However, the GNC particles were almost unobservable surrounding the droplets. 

The quality of English language should be improved further

Reviewer 3 Report

The article entitled “Important Role and Properties of Granular Nanocellulose Particle in

the In Vitro Simulated Gastrointestinal System and Lipid Digestibility and Permeability” describes the granular nanocellulose particles (GNC) from sugarcane bagasse in reducing lipid digestibility and permeability in the in vitro simulated gastrointestinal (GI) system using GNC-olive oil Pickering emulsion. The authors shows that GNC could be utilized as an alternative food additive or supplement in fatty food for weight control due to their inhibiting lipid digestibility and assimilation.

I recommend it is highly suitable for publication in Biomolecules after minor revision. The revisions are,

Introduction

Page number 1, line number 39, the authors should correct the spelling “ac-ids” instead of acids.

Page number 2, line number 75-76, the authors said three types of nanocellulose, but they represent only two types. The authors should add the third type of nanocellulose.

Materials and methods

Page number 4, line number 173-178, The authors should clearly mention the oil water ratio for the emulsion preparation and what type of sonication and amplitude they used to prepare the emulsion.

Round 2

Reviewer 2 Report

The paper could be accepted in present form